# The Effects of Sacubitril/Valsartan on Clinical, Biochemical and Echocardiographic Parameters in Patients with Heart Failure with Reduced Ejection Fraction: The “Hemodynamic Recovery”

**DOI:** 10.3390/jcm8122165

**Published:** 2019-12-06

**Authors:** Giuseppe Romano, Giuseppe Vitale, Laura Ajello, Valentina Agnese, Diego Bellavia, Giuseppa Caccamo, Egle Corrado, Gabriele Di Gesaro, Calogero Falletta, Eluisa La Franca, Chiara Minà, Salvatore Antonio Storniolo, Filippo Maria Sarullo, Francesco Clemenza

**Affiliations:** 1Cardiology Unit and Research Office, Department for the Treatment and Study of Cardiothoracic Diseases and Cardiothoracic Transplantation, Mediterranean Institute for Transplantation and Advanced Specialized Therapies (IRCCS-ISMETT), Via Tricomi 5, 90127 Palermo, Italy; vagnese@ismett.edu (V.A.); dbellavia@ismett.edu (D.B.); gdigesaro@ismett.edu (G.D.G.); cfalletta@ismett.edu (C.F.); elafranca@ismett.edu (E.L.F.); cmina@ismett.edu (C.M.); fclemenza@ismett.edu (F.C.); 2Cardiovascular Rehabilitation Unit, Buccheri La Ferla Fatebenefratelli Hospital, via Messina Marine, 197, 90123 Palermo, Italy; giuseppevit@hotmail.com (G.V.); lajello@ismett.edu (L.A.); caccamo.giuseppa@libero.it (G.C.); sarullo.filippo@fbfpa.it (F.M.S.); 3Cardiology Unit, University Hospital, Policlinico Paolo Giaccone, via del Vespro 129, 90127 Palermo, Italy; egle.corrado@libero.it (E.C.); salvo.storniolo@gmail.com (S.A.S.)

**Keywords:** heart failure, sacubitril/valsartan, neprilysin inhibition, reduced ejection fraction, echocardiography, Nt-ProBNP, hemodynamic, remodeling

## Abstract

Background: Sacubitril/valsartan has been shown to be superior to enalapril in reducing the risks of death and hospitalization for heart failure (HF). However, knowledge of the impact on cardiac performance remains limited. We sought to evaluate the effects of sacubitril/valsartan on clinical, biochemical and echocardiographic parameters in patients with heart failure and reduced ejection fraction (HFrEF). Methods: Sacubitril/valsartan was administered to 205 HFrEF patients. Results: Among 230 patients (mean age 59 ± 10 years, 46% with ischemic heart disease) 205 (89%) completed the study. After a follow-up of 10.49 (2.93 ± 18.44) months, the percentage of patients in New York Heart Association (NYHA) class III changed from 40% to 17% (*p* < 0.001). Median N–Type natriuretic peptide (Nt-proBNP) decreased from 1865 ± 2318 to 1514 ± 2205 pg/mL, (*p* = 0.01). Furosemide dose reduced from 131.3 ± 154.5 to 120 ± 142.5 (*p* = 0.047). Ejection fraction (from 27± 5.9% to 30 ± 7.7% (*p* < 0.001) and E/A ratio (from 1.67 ± 1.21 to 1.42 ± 1.12 (*p* = 0.002)) improved. Moderate to severe mitral regurgitation (from 30.1% to 17.4%; *p* = 0.002) and tricuspid velocity decreased from 2.8 ± 0.55 m/s to 2.64 ± 0.59 m/s (*p* < 0.014). Conclusions: Sacubitril/valsartan induce “hemodynamic recovery” and, consistently with reduction in Nt-proBNP concentrations, improve NYHA class despite diuretic dose reduction.

## 1. Introduction

Liking renin-angiotensin-aldosterone system (RAAS) blockade with natriuretic peptide system enhancement may bear specific therapeutic benefits to patients with heart failure and reduced ejection fraction (HFrEF). The first-in-class angiotensin receptor neprilysin inhibitor (ARNI) sacubitril/valsartan combines the angiotensin II type-1 receptor blocker (ARB) valsartan with the neprilysin inhibitor sacubitril. Sacubitril/valsartan was superior to enalapril in decreasing risks of death and new admission for HF in patients with HFrEF in the Prospective Comparison of ARNI with ACEI to Determine Impact on Global Mortality and Morbidity in Heart Failure (PARADIGM-HF) study [1]. However, the effect of sacubitril/valsartan on cardiac performance in patients with HFrEF remains limited [2,3]. Therefore, in this study, we attempted to assess the effects of sacubitril/valsartan on clinical, biochemical and echocardiographic, parameters in HFrEF patients.

## 2. Experimental Section

Study Design and Patient Selection: The study was conducted in our outpatient HF clinic center, which is mainly focused on HFrEF patients evaluated for heart transplantation, between 1^st^ September, 2017 and 15^th^ January, 2019, and was approved by the Institutional Research Review Board of the Istituto Mediterraneo per i Trapianti e Terapie ad Alta Specializzazione (ISMETT) center in Palermo, Italy. In this prospective observational single center study, sacubitril/valsartan was administered to patients with HFrEF, in addition to recommended therapy [2]. The aim of the study was to evaluate the effects of sacubitril/valsartan on clinical, biochemical and echocardiographic parameters, recorded at baseline and after follow-up.

Patients were included in the study in accordance with the following inclusion criteria: (1)symptomatic heart failure defined as New York Heart Association (NYHA) class II-III;(2)left ventricular ejection fraction (LVEF) below 35% measured by echocardiography;(3)pretreatment with an individual optimal dose of angiotensin-converting enzyme inhibitor (ACE-I) or ARB for at least 6 months;(4)arterial blood pressure ≥100 mmHg;(5)serum potassium (K+) level <5.4 mEq/L. 


Exclusion criteria were as follows: (1)Hospital admission for HF within 90 days before ambulatory evaluation;(2)Myocardial revascularization within 180 days before ambulatory visit;(3)Concomitant implantation of cardiac resynchronization therapy (CRT) and/or percutaneous mitral valve treatment during study follow-up or in the previous 6 months;(4)Presence of congenital heart disease;(5)Severe liver insufficiency (Child–Pugh C);(6)History of angioedema.

All patients provided informed consent for participation, and the protocol was approved by the research ethics committee in accordance with the principles of the Declaration of Helsinki and national regulations. 

Study Procedures: To assess clinical stability, patients were assessed in our outpatient clinic at the enrolment phase (baseline visit). Medical history, physical exam, weight, blood pressure, NYHA class, 12 lead electrocardiogram (ECG), and laboratory analysis comprehensive of biomarkers including N-terminal pro-brain natriuretic peptide (NT-proBNP) were obtained every 1 month to undertake sacubitril/valsartan dose up-titration and then every 6 months. Doses of sacubitril/valsartan were prescribed according to established recommendations [4]. The recommended starting dose was 49/51 mg twice daily. Patients were switched from an ACE-I after a 36 hour washout period. For patients with severe renal impairment (estimated glomerular filtration rate [eGFR] <30 mL/min), moderate liver insufficiency (Child–Pugh B), hypotensive (<110 mmHg), or taking low doses of ACE-I or ARB, the starting dose was 24/26 mg twice-daily. Up-titration was operated every 4 weeks if tolerated by the patient. Furosemide dose modifications were conceded during follow-up. Safety and tolerability assessments were performed, including monitoring and recording of all adverse events and their relationship to the study drug. Two hundred thirty were initially enrolled. After the run-in phase (one month), eight patients discontinued sacubitril/valsartan because of hypotension, four because of worsening renal function and two because of skin erythema: Two hundred and sixteen patients were finally evaluated.

Echocardiography: A standard 2-dimensional and Doppler transthoracic echocardiogram was performed at two time points (baseline assessment and 6 months after the initiation of sacubitril/valsartan) in all patients. All ultrasound examinations were done with a commercially available echocardiographic instrument (Vivid 9 System, Vingmed, General Electric Healthcare and Philips Medical Systems, EPIC, Cary, NC, USA). LVEF and volumes were measured from apical views using the modified biplane Simpson method, as previously described [5]. Volumes and mass were indexed to the body surface area. The right ventricular (RV) longitudinal systolic function was assessed by tricuspid annular plane systolic excursion (TAPSE). Color Doppler was used to qualitatively assess mitral regurgitation (MR) degree. Assessment of diastolic function was made by trans-mitral early (E wave velocity) and late (A wave velocity) Doppler flow waves, E/A ratio, and E deceleration time, and by measuring the early diastolic pulsed wave tissue Doppler (PW-TDI) at the medial and lateral mitral annulus (e’). E/e’ ratio was used as a parameter of LV end-diastolic filling pressure (LVEDP) [6]. Tricuspid regurgitation (TR) velocity was measured in order to estimate systolic pulmonary arterial pressure and inferior vena cava diameter variation as a surrogate for central venous pressure. Images were analyzed offline by two expert investigators blinded to clinical factors as well as drug treatment.

Statistical analysis was executed using SPSS Statistics 25 software (IBM). Continuous variables are defined by mean (SD), or by median and interquartile range, in case of non-normal distribution. Categorical variables were described as number (percentages). Two hundred and five patients were followed-up in our outpatient clinic, and changes from baseline were tested by paired t-test or McNemar test, respectively. A *p*-value <0.05 was considered statistically significant.

## 3. Results

### 3.1. Baseline Evaluations

A total of 216 patients were prospectively enrolled. However, five patients discontinued sacubitril/valsartan because they experienced hypotension, four patients because of acute on chronic HF and two patients had ventricular arrhythmia. Therefore, 205 (89%) patients were included in the final analysis with a median follow-up of 10.49 m (range 2.93–18.44) months. The mean age was 59 ± 10 years, which is younger than general HFrEF population, but consistent with patients usually referred to a transplantation center, 15% females, 46% with ischemic heart disease, 62% with NYHA functional class II and 17% on atrial fibrillation. Baseline characteristics of patients are presented in Table 1.

The mean (SD) of systolic blood pressure was 118.5 ± 15 mm Hg. The median of NT-proBNP levels, eGFR (Modification of Diet in Renal Disease [MDRD] Study) equation dosages, creatinine concentrations and serum potassium at baseline were 1865 ± 2318 pg/mL, 69.4 ± 23.1 mL/min/1.73 m^2^, 1.2 ± 0.35 mg/dL, 4.14 ± 0.44 mEq/L respectively. Beta-blockers, mineralocorticoid receptor antagonist, and furosemide were administered in 96%, 85%, and 88% of patients, respectively. The mean daily furosemide dose was 131.3 ± 154.5 mg. Eighty percent of patient underwent to cardiac defibrillator (ICD) implantation and 25% of patients received CRT device with ICD. The starting dose of sacubitril/valsartan was 24/26 mg twice daily in 77% of patients. The dose of 49/51 mg was administered in 23% of patients. Mean baseline values of LVEF, E/A ratio, left atrial volume index (LAVi), were 27 ± 5.9%, 1.67 ± 1.21 and 54.2 ± 22.6 mL respectively. The percentage of patients with moderate to severe functional MR was 30.1% and the mean baseline values of TR velocity was 2.8 ± 0.55 m/s (Table 2). 

### 3.2. Change in Clinical Characteristics, ARNI dose and Laboratory Data 

After a median follow-up of 10.49 months (2.93 ± 18.44 months), percentage of patients HYHA class II increase from 60% to 73% and the number of patients in NYHA class III decrease from 40% to 17% (*p* < 0.001). 

Systolic and diastolic blood pressure decreased with treatment (*p* = 0.009 and *p* < 0.001, respectively). The dose of sacubitril/valsartan 49/51 mg twice daily was administered in 34% of patients. In 39% of patients, the initial dosage of 24/26 mg twice daily was maintained. The dose was up titrated until 97/103 mg twice daily only in the 27% of patients, because of symptomatic hypotension. The median furosemide dose decreased from 131.3 ± 154.5 mg at baseline to 120 ± 142.5 mg after follow-up (*p* = 0.047), see Table 2. Initiation and titration of sacubitril-valsartan was associated with a reduction in NT-proBNP concentration (1514 ± 2205 pg/mL; *p* =0.01). We observed significant changes, but not clinically relevant, in eGFR (65.3 ± 23.2 mL/min/1.73 m^2^; *p =* 0.012). Only two patients with eGFR <30 mL/min/1.73 m^2^ were included and consequently we did not perform subgroup analysis 

In these two patients, Sacubitril/Valsartan was less titrated compared to patients with eGFR ≥60 mL/min/1.73 m^2^ and moreover, they did not experience eGFR worsening during follow-up. 

No variation in creatinine concentrations and in serum potassium (1.31 ± 0.57 mg/mL; *p = 0.052*) (4.17 ± 0.44 mEq/L, *p* = 0.611) were founded, see Table 2. 

### 3.3. Change in Echocardiographic Measurements.

Patients exhibited a mild but significant improvement in LVEF (30 ± 7.7%; *p* = 0.001). The changes in the E/A-wave ratio from baseline to follow-up were (1.42 ± 1.12; *p* = 0.002), on the contrary there was no significant change in E/e’ (from 14.79 ± 6.10 to 13.85 ± 6.09; *p* = 0.194). Treatment with sacubitril-valsartan was also associated with significant reduction of the percentage of patients with moderate to severe MR (from 30.1% to 17.4%, *p* = 0.002). In addition, TR velocity decrease from 2.8 ± 0.55 m/s to 2.64 ± 0.59 m/s (*p* < 0.014), (Table 2). 

### 3.4. Safety

During follow-up, five (2%) patients discontinued sacubitril/valsartan because they experienced hypotension and four (2%) patients because of acute on chronic HF. In two (1%) patients, worsening renal function was observed. 

### 3.5. Outcomes

During follow-up, no patients died. In the group of ischemic cardiomyopathy, we observed one hospital admission because of acute on chronic HF and one admission because ventricular arrhythmia. 

Concerning non ischemic cardiomyopathy, we found one acute on chronic hospitalization.

## 4. Discussion

This prospective observational study of patients with HFrEF showed that switching to sacubitril/valsartan may generate “hemodynamic recovery” by reducing left ventricular filling pressure, MR and finally pulmonary artery systolic pressure. This hemodynamic effect in association with the reduction of Nt-proBNP may ameliorate functional class capacity and identify patients in which diuretic withdrawal could be safely performed (Figure 1). 

In this study, we evaluated the effect of switching to sacubitril/valsartan therapy in HFrEF patients through a multiparametric approach, that is NT-proBNP levels, echocardiography, and NYHA Class and all collected data were used to test the hypothesis that sacubitril/valsartan may confer an early comprehensive and global benefit to HFrEF patients. 

We decided to exclude patients with recent admission because of acute on chronic HF so to select stable patients on firm medical regimen. In addition to their vasodilatatory, natriuretic, and diuretic effects, natriuretic peptides inhibit the RAAS, sympathetic nervous system, and consequent release of antidiuretic hormone, improve myocardial relaxation and vagal tone, and have antifibrotic and antihypertrophic properties [7,8]. Mechanistically, sacubitril is implicated in attenuating cardiomyocyte cell death, hypertrophy, and impaired myocyte contractility [9]. Based on these preclinical and mechanistic evaluations of sacubitril, the incremental beneficial effect systolic and diastolic function might seem more intuitive than expected. However, prospective data regarding sacubitril-valsartan and cardiac remodeling are limited: Martens and colleagues [2] reported a 5% mean improvement in LVEF after a follow-up period of 4 months. The recent PROVE-HF study [3] adds information regarding associations between ARNI therapy, change in NT-proBNP, and cardiac remodeling. Reduction in NT-proBNP following treatment with sacubitril-valsartan was associated with an increase in LVEF, and reductions in indexed LV and LA volumes as well as E/e′ ratio. In line with these findings we found a mild but significant improvement in cardiac function measured by LVEF, confirming the potential LV reverse remodeling effect mediated by sacubitril/valsartan but neither significant reductions in LV and LA volume nor in E/e’ ratio was noticed. In our opinion this inconsistency can be explained by the fact that in our study patients had significantly dilated ventricular and atrial volumes and higher NT-proBNP values, which, in conclusion, would suggest a more advanced HF disease than that of the PROVE HF study [3], needing more time to observe reverse remodeling.

Moreover, consistently with previous study [2] we reported a reduction in E/A ratio as well as improvement of MR severity [10]. Both are important prognostic measures, reflecting the magnitude and chronicity of elevated cardiac filling pressures, LV negative remodeling and fluid congestion. 

Although qualitatively assessed, the MR grading reduction it has been further confirmed and proven by the reduction in TR velocity that means pulmonary artery systolic pressure lowering. These data are unique and fascinating and are linked with the significant improvement in NYHA class observed in our population.

Coherently with echocardiographic measurements, neprilysin inhibition mediated by sacubitril acutely amplified the hemodynamic effects of natriuretic peptides determining natriuresis and vasodilation [11,12] which resulted in decreased neurohormonal activation as our data have demonstrated by NT-proBNP concentrations abatement at follow-up. 

In facts, reduction in NT-proBNP concentration was strongly associated with outcomes in PARADIGM-HF [1]. On the other hand, studies have suggested that a lack of NT-proBNP reduction after therapy for HFrEF is associated with worse left ventricular size and function [13,14].

Our results suggest that patients with NT-proBNP reduction following ARNI initiation are likely to experience reverse cardiac remodeling. 

Improving in filling pressure, MR degree and pulmonary pressure in tandem with a small yet significant improvement in EF, that is “hemodynamic recovery”, effectively improved NYHA class and exertional dyspnea. 

In a recent metanalysis of twenty studies enrolling 10,175 patients, ARNI improved functional capacity in patients with HFrEF, including increasing NYHA class and 6 minute walking distance. Moreover, ARNI outperformed angiotensin-converting enzyme inhibitors/angiotensin receptor blockers in terms of cardiac reverse remodeling with striking changes in left ventricular EF, diameter, and volume [15].

Confirming these data, we found a reduction of percentage of patients in NYHA class III and an increasing number of patients in NYHA class I and II at follow-up (Figure 1). 

These data are in line with our previously published results showing a significant improvement in well-known surrogates of cardiac performance such as peak VO_2_ and O_2_ pulse as well as others main prognostic-relevant CPET parameters after initiation of sacubitril/valsartan [16]. 

Furthermore, this “hemodynamic recovery” in association with Nt-proBNP concentration reduction, could lead to identify patients in which diuretic withdrawal strategy can be safely undertaken [17]. As we founded in our study reducing the mean diuretic dose, allows avoiding a significant deterioration of renal function, [18] and electrolyte imbalance. 

Interestingly treatment with sacubitril/valsartan was associated with more loop diuretic dose reductions and fewer dose increases compared with enalapril in the PARADIGM-HF study [19], suggesting that treatment with sacubitril/valsartan may reduce the relative requirement for loop diuretics in patients with heart failure with reduced ejection fraction. The reduced relative need for diuretics in patients treated with sacubitril/valsartan may potentially be secondary to the natriuretic effects of sacubitril or the presumed improvement in hemodynamics that may occur with sacubitril/valsartan. 

Loop diuretic use has been associated in prior studies with worse outcomes in heart failure. Several mechanisms have been proposed by which loop diuretics may increase risk of mortality: neurohormonal activation electrolytes depletion, serious cardiac arrhythmias [20,21], as well as an increased risk of cardio-renal syndrome [22] have all been reported in the literature. For this reason, as we have already demonstrated [17], diuretic therapy can and should be suspended in well-selected, asymptomatic, patients with HFrEF after adequate therapeutic neuro-hormonal modulation to preserve renal function. Consistently with Damman et al. [23] we have observed a slight worsening in eGFR that anyhow did not reclassify patients in terms of chronic kidney disease severity and was not associated to any change in creatinine or serum potassium levels. On the contrary, Spannella et al. [24] showed an eGFR improvement in patients on Sacubitril/Valsartan. In our opinion, this inconsistency can be explained by significantly higher dosages of MRA and loop diuretics taken by our patients as compared to the heretofore mentioned study. 

This study has a number of limitations. Firstly, the study was not randomized. However, prospective longitudinal studies with multiple blinded assessors are a well-accepted design for evaluating echocardiographic and cardiopulmonary changes. Secondly, an important limitation of this study is the relatively small sample size and lack of a control group. Our study did not show significant reversal in EDVi, although other study observed significant decrease in MR degree severity and in LV volumes [Martens] after switching to sacubitril/valsartan. 

Echocardiography is a standard imaging method for the evaluation of MR, but it is not as accurate as cardiac MRI for measurement of LV volumes. 

However, this study enrolled 80% of patients with ICD and 25% with CRT hampering optimal imaging acquisition for magnetic resonance imaging. 

Our choice of echocardiography as the primary imaging tool could be the limitation for assessment of LV remodeling. 

In addition, some echocardiographic parameters useful in the assessment of reverse remodeling (LV mass index; strain analysis) were not performed. 

MR was visually assessed because quantitative assessment of functional MR could be unreliable: the PISA method is limited by its radius which is frequently not constant, and the geometry of the PISA varies (ellipsoidal shape) underestimating the degree of functional MR. 

Vena contracta was not measured as well because intermediate values are not accurate at distinguishing moderate from mild or severe MR (large overlap); they usually require the use of another method for confirmation.

Furthermore, the magnitude of reverse remodeling associated with the reduction in MR entity might have been substantially underestimated with shorter follow-up.

## 5. Conclusions

In summary, our findings are strongly suggestive of “hemodynamic recovery” in which a modulation of neurohormonal activation determined by sacubitril/valsartan may lead to a hemodynamic effect that may impact cardiac hemodynamic and in association with Nt-proBNP concentration abatement could lead to a ameliorate NYHA class and reduce diuretics administration and consequently to preserve renal function.

## Figures and Tables

**Figure 1 jcm-08-02165-f001:**
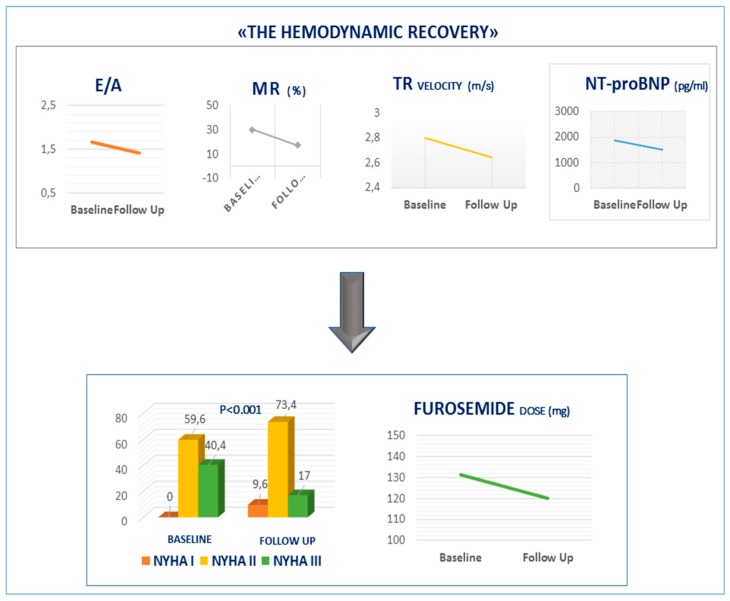
Hemodynamic recovery. Sacubitril/valsartan reduced E/A ratio, MR, TR velocity and Nt-ProBNP concentration. This hemodynamic effect ameliorates the NYHA class and reduce diuretic dose at follow-up. MR, mitral regurgitation from moderate to severe grade; E/A: peak e-wave velocity/ peak a-wave velocity ratio; TR velocity: tricuspid regurgitation peak velocity.

**Table 1 jcm-08-02165-t001:** Baseline characteristics of patients.

Patients Characteristics	N (%)
Patients	205
Age (mean ± SD)	59 ± 10
Female sex	31 (15)
BSA (mean ± SD)	2 ± 0.2
ETIOLOGY	
Ischemic	95 (46)
Non Ischemic	110 (54)
NYHA	
II	128 (62)
III	77 (38)
COMORBIDITY	
hypertension	90 (45)
Diabetes	63 (32)
Atrial fibrillation	35 (17)
COPD	7 (3)
MEDICAL THERAPY	
FUROSEMIDE	180 (88)
MRA	174 (85)
ACE- I /ARB	100 (205)
β-BLOCKERS	197 (96)
IVABRADINE	37 (18)
ELECTRICAL THERAPY	
ICD	164 (80)
CRT	51 (25)

Values are mean ± standard deviation. BSA, Body surface area; NYHA, New York Heart Association; COPD, chronic obstructive pulmonary disease; MRA, mineralocorticoid receptor antagonist; ACE-I, angiotensin-converting enzyme inhibitor; ARB, angiotensin receptor blocker; ICD, intracardiac defibrillator; CRT, cardiac resynchronization therapy.

**Table 2 jcm-08-02165-t002:** Changes in clinical, sacubitril/valsartan dose, biochemical and echocardiographic parameters.

	Baseline	Follow-up	*p* Value
SBP (mmHg)	118.5 ± 15	115.4 ± 16.9	0.042
DBP (mmHg)	73 ± 10.3	67.5 ± 9.3	<0.001
NT-proBNP (pg/mL)	1865 ± 2318	1514 ± 2205	0.01
Creatinine (mg/dL)	1.2 ± 0.35	1.31 ± 0.57	0.052
eGFR (mL/min/1.73 m^2^)	69.4 ± 23.1	65.3 ± 23.2	0.012
potassium (mEq/L)	4.14 ± 0.44	4.17 ± 0.44	0.611
Furosemide dose (mg)	131.3 ± 154.5	120 ± 142.5	0.047
SACUBITRIL/VALSARTAN			
24/26 (mg/bid)	77	39	
49/51 (mg/bid)	23	34	
97/103 (mg/bid)	0	27	
FE (%)	27 ± 5.9	30 ± 7.7	<0.001
EDVi (mL/m^2^)	120.5 ± 31.4	120.7 ± 33	0.932
MR mod/sev (%)	30.1	17.4	0.002
E/A	1.67 ± 1.21	1.42 ± 1.12	0.002
E/e’	14.79 ± 6.10	13.85 ± 6.09	0.194
LAVi (mL/m_2_)	54.2 ± 22.6	52.4 ± 19.1	0.202
TR velocity (m/s)	2.8 ± 0.55	2.64 ± 0.59	0.014
TAPSE (mm)	19.03 ± 4.55	19.28 ± 3.62	0.472

SBP: Systolic blood pressure; DBP: Diastolic blood pressure; Nt-proBNP, N-terminal pro–B-type natriuretic peptide. eGFR, estimated glomerular filtration rate; EF, ejection fraction; EDVi, endiastolic volume index.; MR, mitral regurgitation from moderate to severe grade; E/A: peak e-wave velocity/peak a-wave velocity ratio; E/e’ peak: e-wave velocity divided by mitral annular e’ velocity (average) ratio; LAV-i, left atrial volume index; RA, right atrium; TR velocity: tricuspid regurgitation peak velocity; TAPSE, tricuspid annular plane systolic excursion.

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
