# Peer review of "The Effects of Sacubitril/Valsartan on Clinical, Biochemical and Echocardiographic Parameters in Patients with Heart Failure with Reduced Ejection Fraction: The “Hemodynamic Recovery”"

_jcm, 2019, doi:10.3390/jcm8122165_

Round 1
Reviewer 1 Report
Romano G. and co. presented a prospective study that investigated the effects of sacubitril/valsartan on clinical, bioumoral and echocardiographic parameters in patients with heart failure with reduced ejection fraction. A collective consisted initially of 216 patients. Change in Clinical Characteristics involving NYHA class and a decrease of furosemide dose during follow-up-time as well as a change in echocardiographic measurements and Safety of sacubitril/valsartan were observed. Several limitations were described and could be raised.
Comment on first Section (Introduction):
The introduction has several weaknesses. Martens and co. reported that sacubitril/valsartan improved a cardiac performance involving a diastolic function as well as echocardiographic changes involving E/A-wave within the meaning a reverse remodeling response. These aspects have been poorly studied up to now but doesn’t remain unknown.
Comment on second section (Experimental Section):
The mean age of selected patients is 59 years old. This collective doesn’t reflect the reality of heart failure patients that usually elderly than 59 years old. The exclusion of HF hospitalization within 90 days before ambulatory evaluation; notes that selected patients in this study are healthier than the most HF patients and have a better hemodynamic.
Comment on third section (Results):
Please provide more data about clinical outcomes, depending on HF etiology (in this study 46 % ischemic and 54% non-ischemic). A median follow-up time with 10.49 months is too short to ensure or maintain that sacubitril/valsartan improves hamodynamic performance. The dose of sacubitril/valsartan increased during follow-up from 24/26 mg/dl to 97/103 mg/dl only in 27% of patients, why?
Comment on fourth section (Discussion):
I think, it is important to discuss a significant change in eGFR and compare with other studies (Renal effects of Sacubitril/Valsartan in heart failure with reduced ejection fraction: a real life 1-year follow-up study, Spannella F et al) ( Renal Effects and Associated Outcomes During Angiotensin-Neprilysin Inhibition in Heart Failure, Damman K et al).
This study requires English language editing. For example, you wrote
Line 105 sistolic à systolic Table 1 Pazientsà Patients
Author Response
Comment on first Section (Introduction):
The introduction has several weaknesses. Martens and co. reported that sacubitril/valsartan improved a cardiac performance involving a diastolic function as well as echocardiographic changes involving E/A-wave within the meaning a reverse remodeling response. These aspects have been poorly studied up to now but doesn’t remain unknown.
Thank you for the comment. We agree with the reviewer that the impact of sacubritil/valsaratan on LV performance is not totally unknown and we have modified the text accordingly as follows:
“However the effect of sacubitril/valsartan on cardiac performance in patients with HFrEF remains limited [Martens]. Therefore, in this study, we sought to evaluate the effects of sacubitril/valsartan on clinical, biochemical, echocardiographic, parameters in HFrEF patients.”
Comment on second section (Experimental Section):
The mean age of selected patients is 59 years old. This collective doesn’t reflect the reality of heart failure patients that usually elderly than 59 years old. The exclusion of HF hospitalization within 90 days before ambulatory evaluation; notes that selected patients in this study are healthier than the most HF patients and have a better hemodynamic.
Indeed patients enrolled in our study were younger than comparable published studies on heart failure. Although they seem in better health with stable hemodynamic balance, we have enrolled a population of pretty stable outpatients with HFrEF. Still these patients were considered having a quite severe LV dysfunction, to justify referral to our heart transplant center. We have modified the text accordingly as follows:
“We decided to exclude patients with recent admission because of acute on chronic HF so to select stable patients on firm medical regimen.” “The mean age was 59 ± 10 years, that is younger than general HFrEF population, but consistent with patients usually referred to a transplantation center”
Comment on third section (Results):
Please provide more data about clinical outcomes, depending on HF etiology (in this study 46 % ischemic and 54% non-ischemic). A median follow-up time with 10.49 months is too short to ensure or maintain that sacubitril/valsartan improves hamodynamic performance. The dose of sacubitril/valsartan increased during follow-up from 24/26 mg/dl to 97/103 mg/dl only in 27% of patients, why?
Thank you for the comment. We have modified the text accordingly as follows:
“During follow up no patients died. In group o f ischemic cardiomyopathy we observed 1 hospital admission because acute on crhonic HF and 1 admission because ventricular arrhthmya. Corcerning non ischemic cardiomyopathy we found 1 acute on chronic hospitalization The dose was up titrated until 97/103 mg twice daily only in the 27% of patients, because of symptomatic hypotension”.
Comment on fourth section (Discussion):
I think, it is important to discuss a significant change in eGFR and compare with other studies (Renal effects of Sacubitril/Valsartan in heart failure with reduced ejection fraction: a real life 1-year follow-up study, Spannella F et al) ( Renal Effects and Associated Outcomes During Angiotensin-Neprilysin Inhibition in Heart Failure, Damman K et al).
We would like to thank the reviewer for the comment. We have corrected our statement and modified the text accordingly as follows:
“Consistently with Damman et al. we have observed a slight worsening in eGFR that anyhow did not reclassify patients in terms of chronic kifdney disease severity and was not associated to any change in creatinine or serum potassium levels. On the contrary, Spannella et al. showed an eGFR improvement in patients on Sacubritil/Valsartan. In our opinion this inconsistency can be explained by significantly higher dosages of MRA and loop diuretics taken by our patients as compared to the heretofore mentioned study.”
This study requires English language editing. For example, you wrote
Line 105 sistolic à systolic Table 1 Pazientsà Patient
The English language has been edited and the word in Table 1 has been modified as requested
Reviewer 2 Report
The authors describe in a prospective cohort of 205 HFrEF patients treated with sacubitril/valsartan, a clinical improvement, lowering of filling pressures, a decrease in MR, a decrease in Nt-proBNP and an improvement in EF.
The study is well performed and methodologically correct. The document should reviewed at the linguistic level by a native speaker. The word “bioumoral” in the titel should be replaced and there are grammatical errors which cannot be accepted in the conclusion of the abstract as well as in the conclusion of the manuscript.
As regards content, I do have some specific comments.
The findings are interesting and partially concordant with previous studies. The echocardiographic data add further evidence how treatment with sacubitril/valsartan improves outcome. In contrast to Martens et al. no change in LVEDVi is found, only a decrease in EF and improvement in hemodynamics. Therefore the conclusion of sacubitril/valsartan induced reverse remodeling (defined by change in LV volume) cannot be made based on the findings in this study. An improvement of EF can also be result of a decrease in afterload; so this finding is insufficient to conclude that sacubitril/valsartan induces reverse remodeling and the statement "hemodynamic reverse remodeling" should be abandoned.
The authors also included patients with severe renal insufficiency with eGFR < 30ml/min. I would be interested in the number of patients with severe renal insufficiency, the tolerability in this subgroup and the lower cut-off for inclusion in the study.
The authors mention that this is the first study report which shows a reduction in E/A ratio as well as a reduction in MR severity. However this is already described by Martens et al (ref 8) and in the PRIME-HF study by Kang Circulation. 2019 Mar 12;139(11):1354-1365. doi: 10.1161/CIRCULATIONAHA.118.037077.
Author Response
The study is well performed and methodologically correct. The document should reviewed at the linguistic level by a native speaker. The word “bioumoral” in the title should be replaced and there are grammatical errors which cannot be accepted in the conclusion of the abstract as well as in the conclusion of the manuscript.
Thank you for the comment the word “bioumoral” has been taken out and replaced with “biochemical”.
As regards content, I do have some specific comments.
The findings are interesting and partially concordant with previous studies. The echocardiographic data add further evidence how treatment with sacubitril/valsartan improves outcome. In contrast to Martens et al. no change in LVEDVi is found, only a decrease in EF and improvement in hemodynamics. Therefore the conclusion of sacubitril/valsartan induced reverse remodeling (defined by change in LV volume) cannot be made based on the findings in this study. An improvement of EF can also be result of a decrease in afterload; so this finding is insufficient to conclude that sacubitril/valsartan induces reverse remodeling and the statement "hemodynamic reverse remodeling" should be abandoned.
Thank you for the comment. We have in facts taken out the statement "hemodynamic reverse remodeling" as requested and replaced it with the more correct “hemodynamic reversion”
The authors also included patients with severe renal insufficiency with eGFR < 30ml/min. I would be interested in the number of patients with severe renal insufficiency, the tolerability in this subgroup and the lower cut-off for inclusion in the study.
Actually, only 2 patients among the recruited population had an eGFR < 30 ml/min. Because of such low number of patients no subgroup analysis was feasible. We have specified this in the following sentence:
Only two patients with eGFR < 30 ml/min/1.73 m2 were included and consequently we did not perfomed subgroup analysis
In this two patients Sacubitril/Valsartan was less titrated compared to patients with eGFR ≥ 60 ml/min/1.73 m2 and moreover, they did not experience eGFR worsening during follow up.
The authors mention that this is the first study report which shows a reduction in E/A ratio as well as a reduction in MR severity. However this is already described by Martens et al (ref 8) and in the PRIME-HF study by Kang Circulation. 2019 Mar 12;139(11):1354-1365. doi: 10.1161/CIRCULATIONAHA.118.037077.
We would like to thank the reviewer for the comment. We have corrected our statement and modified the text accordingly as follows:
“Moreover, consistently with previous study [Martens et al] we reported a reduction in E/A ratio as well as improvement of MR severity [PRIME HF]. Both are important prognostic measures, reflecting the magnitude and chronicity of elevated cardiac filling pressures, LV negative remodeling and fluid congestion.
Although qualitatively assessed, the MR grading reduction it has been further confirmed and proven by the reduction in TR velocity and in the pulmonary artery systolic pressure. To the best of our knowledge this is the first study in which the ARNI benefit was observed not only in inducing, E/A ratio and MR severity reduction but also in pulmonary pressure lowering. This data are unique and fascinating and are linked with the significant improvement in NYHA class observed in our population. “
Reviewer 3 Report
The authors have executed a well designed study and acknowledge the limitations in their discussion. Several comments would improve the manuscript in my opinion:
Several echo measurements could be included in the table: LV mass index; e' (lateral and medial); strain if obtained; and LV wall thickness. The MR severity is one of the novel findings of the study. That being said color Doppler alone could be a relatively subjective method to measure severity. Did the authors have other measures, such as ERO, regurgitant fraction, or vena contracta to quantitate the degree of MR? This would greatly improve the evidence for this finding Along the same lines, the authors were blinded to "clinical factors"? Were they also blinded to the time course? IN other words did they know whether the echo was the baseline or 6 month echo? This will be important to discuss. The Figure could be improved in several ways: a. labeling the y axes with units; b. in the HF class sub-figure, denoting the P values. Finally, the findings must be interpreted in comparison with the published study in JAMA which already demonstrated improved LVEF with sacubitril/valsartan. It would be very helpful for the authors to discuss and interpret: a) why MR improved in this cohort but not in that study; b) why end diastolic volume improved in the published study but not in the authors study. Any discussion of and descriptions of how the patient populations differ or are similar between these two studies will also be useful.Author Response
Several echo measurements could be included in the table: LV mass index; e' (lateral and medial); strain if obtained; and LV wall thickness. The MR severity is one of the novel findings of the study. That being said color Doppler alone could be a relatively subjective method to measure severity. Did the authors have other measures, such as ERO, regurgitant fraction, or vena contracta to quantitate the degree of MR? This would greatly improve the evidence for this finding.
Unfortunately, we have not performed any quantitative assessment on MR grading in our study population, so that EROA, Regurgitant Fraction or even Vena Contracta are not available. We have specified these significant limitations of the study as follows:
“In addition some echocardiographic parameters useful in the assessment of reverse remodeling (LV mass index; strain analysis) were not performed.
MR was visually assessed because quantitative assessment of functional MR could be unreliable: the PISA method is limited by its radius which is frequently not constant and the geometry of the PISA varies (ellipsoidal shape) underestimating the degree of functional MR.
Vena contracta was not measured as well because intermediate values are not accurate at distinguishing moderate from mild or severe MR (large overlap); they usually require the use of another method for confirmation.”
Along the same lines, the authors were blinded to "clinical factors"? Were they also blinded to the time course? IN other words did they know whether the echo was the baseline or 6 month echo? This will be important to discuss.
Thank you for the comment. In the paper we stated: “Images were analyzed offline by two expert investigators blinded to clinical factors as well as drug treatment”.
The Figure could be improved in several ways: a. labeling the y axes with units; b. in the HF class sub-figure, denoting the P values.
Figure has been modified as requested:
Finally, the findings must be interpreted in comparison with the published study in JAMA which already demonstrated improved LVEF with sacubitril/valsartan. It would be very helpful for the authors to discuss and interpret: a) why MR improved in this cohort but not in that study; b) why end diastolic volume improved in the published study but not in the authors study. Any discussion of and descriptions of how the patient populations differ or are similar between these two studies will also be useful.
We would like to thank the reviewer for the comment. Unfortunately It seems that in PROVE HF the echocardiographic evaluation of MR was not assessed (a).
In addition we have corrected our statement and modified the text accordingly as follows (b):
“In our opinion this inconsistency can be explained by the fact that in our study patients had significantly dilated ventricular and atrial volumes and higher NT-proBNP values, which in conclusion would suggest a more advanced HF disease than that of the PROVE HF study needing more time to observe reverse remodeling”.